# Data-driven scenario-based model projections and management of the May 2021 COVID-19 resurgence in India

Edwin Michael[1]*, Ken Newcomb[1], Anuj Mubayi[2,3]

**1** Global Health Infectious Disease Research, University of South Florida, Tampa, FL, United States of America, **2** PRECISIONheor, Los Angeles, CA, United States of America, **3** Center for Collaborative Studies in Mathematical Biology, Illinois State University, Normal, IL, United States of America

* emichael443@usf.edu

## Abstract

The resurgence of the May 2021 COVID-19 wave in India not only pointed to the explosive speed with which SARS-CoV-2 can spread in vulnerable populations if unchecked, but also to the gross misreading of the status of the pandemic when decisions to reopen the economy were made in March 2021. In this combined modelling and scenario-based analysis, we isolated the population and policy-related factors underlying the May 2021 viral resurgence by projecting the growth and magnitude of the health impact and demand for hospital care that would have arisen if the spread was not impeded, and by evaluating the intervention options best able to curb the observed rapidly developing contagion. We show that only by immediately re-introducing a moderately high level of social mitigation over a medium-term period alongside a swift ramping up of vaccinations could the country be able to contain and ultimately end the pandemic safely. We also show that delaying the delivery of the 2nd dose of the Astra Zeneca vaccine, as proposed by the Government of India, would have had only slightly more deleterious impacts, supporting the government's decision to vaccinate a greater fraction of the population with at least a single dose as rapidly as possible. Our projections of the scale of the virus resurgence based on the observed May 2021 growth in cases and impacts of intervention scenarios to control the wave, along with the diverse range of variable control actions taken by state authorities, also exemplify the importance of shifting from the use of science and knowledge in an ad hoc reactive fashion to a more effective proactive strategy for assessing and managing the risk of fast-changing hazards, like a pandemic. We show that epidemic models parameterized with data can be used in combination with plausible intervention scenarios to enable such policy-making.

## Introduction

The devastating second wave of COVID-19 that unfolded in India during April and May 2021 [1,2] demonstrates, on the one hand, the explosive power and speed with which disease outbreaks caused by extremely contagious airborne viral pathogens can spread in populations and

**Data Availability Statement:** The computation system, data and model parameters are available at: https://github.com/EdwinMichaelLab/COVID-SEIR-India.

**Funding:** The authors received no specific funding for this work.

**Competing interests:** The authors have declared that no competing interests exist.

overwhelm national health systems if unchecked. It also epitomizes the challenges to governmental decision making in responding to a complex infectious disease that is marked by major transmission uncertainties, surveillance challenges, and asymmetry in the risks produced (e.g. trade-offs between multidimensional impacts on health, economy and society at large) that make it difficult to formulate effective policies [3–5]. As has been pointed out, these difficulties are further exacerbated when political and economic exigencies favor a particular reading of the "evidence" to support societal reopening decisions in the midst of ongoing pandemics [1,6].

The COVID-19 emergency in India that took place in May 2021 also focused attention on the important need to continuously evaluate and forecast the likely future paths that can be taken by a pandemic, so that sound adaptive action can be taken to manage an outbreak as transmission conditions change through time. Two key pieces of dynamic information are paramount for managing a pandemic wave: viz. (i) what the likely impacts of the surge might be in terms of the development of the pandemic, healthcare demand, and fatalities over time, and (ii) how best to curb the ongoing transmission to reduce these health burdens rapidly [7–10]. These specifically include, on the one hand, data on how quickly the pandemic is growing, when it will peak, its overall magnitude and health impacts, and how long it may last, while results of intervention modelling analyses to inform policies are also required secondly to inform on how best to respond effectively to control potentially catastrophic outcomes [9,10].

Epidemic models offer an important informational tool for making both these situational and pandemic response assessments in the outbreak emergency management cycle [8,10,11]. For example, these models have been instrumental in identifying the best mix of traditional non-pharmaceutical epidemic control measures like social distancing, mask wearing, and isolation of individuals, to curb viral spread appropriately during the previous waves of the pandemic, and increasingly for assessing the future course of the contagion in vaccinated populations [12–16]. In addition to facilitating epidemic forecasts, an important role these tools can play is also in improving the public's and policy makers' understanding of the factors involved in the changing risk status of a population, and how such changes in turn may drive future pandemic growth [8,17].

To be useful for policy making, however, it is critical that models are able to reliably capture the effects of the rapidly changing social contexts in which real world epidemics evolve, and have the means for incorporating feedback from these changing conditions sufficiently rapidly to correct for modelling errors and altered circumstances [18,19]. They also need to take full advantage of the diverse data sources and surveillance streams that become available during the course of an epidemic so that changes in transmission as well as intervention conditions are addressed sufficiently to allow reliable forecasts [20]. Data-model assimilation, in which information regarding the extant transmission processes that are embedded in observation data is used to update the underlying dynamic principles represented by the structure and parameters of a model, offers a means to overcome the above challenges by providing near-term forecasts which are better than could be obtained with just data or the model alone [16–19,21].

Here, our objectives are threefold. First, to project the size and likely path the May 2021 viral resurgence in India could have taken if new control options were not enacted; second, to facilitate insights into the underlying factors that drove the resurgence of cases in May 2021; and third, to use the results of scenario-based intervention modelling in order to evaluate the measures required to achieve control of the resurgence. We extended our previous data-driven socioecological SEIR-based COVID-19 mathematical model to include the dynamics of imperfect vaccines, and applied it to both the case and vaccination rate data reported for India in order to address these objectives. As of May 8, 2021, India had yet to impose a national

lockdown, so social mitigation policies were left largely to individual state governments. The measures taken by the Indian states, however, varied significantly ranging from imposition of total societal lockdowns of variable durations by half of the states to partial shutdowns carried out in other states [22]. Many large states (Uttar Pradesh, Maharashtra, West Bengal) also imposed social restrictions that lasted well into July and August, while other states (Bihar, Madhya Pradesh) appear to have reopened earlier in late June or July [23]. This diversity in the state response—summarized more fully in Supporting Information S1 Table—has made it difficult to evaluate how India managed to curb and achieve control of the May 2021 pandemic crisis. We addressed this topic in this paper by comparing the results arising from our scenario-based intervention modelling exercise against data on the course of the wave that was observed in India subsequently from May 2021 to end Dec 2021, which enable us to shed light on the management responses that allowed curbing and control of the resurgence in the country. We also used these simulations in conjunction with outcomes of the policies that were carried out in the country to assess the relative merits of using knowledge reactively, as carried out in India, versus employing it proactively based on projections of future epidemic states for bringing about the effective control of a disease contagion.

## Materials and methods

### Extended SEIR model

Our previous data-driven socio-ecological SEIR-based COVID-19 model [24] that facilitated incorporation of the effects of changes in social mitigation measures was extended to include the dynamics of imperfect vaccines in order to perform the present simulations. Briefly, the model simulates the course of the pandemic in a population through the adaptive rate of movement of individuals through various discreet compartments, including different infection and symptomatic categories as well as immune, vaccination and death classes as both a function of time and as a result of temporal changes in the social mitigation measures applied or followed by the population. We also assumed that the modelled population is closed, and that the population size remains constant over the duration of the simulations reported here. The coupled differential equations governing the evolution of the system, and the model code used to perform the simulations are available at: https://github.com/EdwinMichaelLab/COVID-SEIR-India.

### Sequential model calibration

Calibration of the model to capture the transmission conditions of India was performed by fitting the SEIR model sequentially to daily confirmed case, death, and vaccination data assembled from the start of the epidemic until May 5th 2021, as provided by the The Coronavirus App [25]. A 7-day moving average is applied to the daily confirmed case and death data to smooth out fluctuations due to COVID-19 reporting inconsistencies. A sequential Monte Carlo-based ensemble approach was used for carrying out the updating of the model by sampling 50,000 initial parameter vectors initially from prior distributions assigned to the values of each parameter for every 10-day block of data [14,24]. An ensemble of 250 best-fitting parameter vectors, based on a Normalized Root Mean Square Error (NRMSE) between predicted and observed case and death data [21], is then selected for describing these 10-day segments of data. Updating of the parameters is then accomplished by using the best-fitting ensemble of parameter posteriors as priors for the next 10-day block, and the fitting process is repeated. In addition, 50% of parameter vectors is drawn from the initial prior distribution to avoid parameter depletion during each updating episode [26]. The strength of social restrictive measures imposed by authorities to limit contacts is captured through the estimation of a

scaling factor, *d*, which is in turn multiplied by the transmission rate, *beta*, to obtain the population-level transmission intensity operational at any given time in a population. This factor accounts for the effects of stay-at-home, shut down of public places, mask wearing, reductions in mobility and mixing, and any other deviations from the normal social behavior of a population prior to the epidemic. To estimate *beta* and *d*, we first obtained values for plausible priors from the literature [27–31] and used data during the sequential Bayesian calibration process to quantify their posterior parameter values at any given time [24]. All prior and posterior fitted parameter values for the best-fit models calibrated to data to May 5th 2021 are given in the Table provided at: https://github.com/EdwinMichaelLab/COVID-SEIR-India.

### Forecasting intervention effects

The ensemble of best-fitting models obtained from the sequential model calibrations carried out using data to May 5th 2021, as described above, was used to forward forecast the impacts of different plausible intervention scenarios. Such scenario-based modelling has been shown to constitute a powerful tool not only for providing projections of the future state of a dynamical system in the absence of any additional interventions (the "reference" or "baseline" case) but also for illustrating how alternative policies may be used to achieve a desired system future [32].The impact of vaccination scenarios is simulated by directly moving the proportion of the population that is reported to have been vaccinated in the country over a 10-day time interval [25] from the susceptible class to the vaccine (1st dose) class. Individuals are then moved from the vaccine to the booster (2nd dose booster) class at a daily rate approximating a 6-week interval between vaccine doses. We initially assume a 1st dose efficacy against acquisition of infection (the degree protection) to be 67% while this is raised to 82% following the 2nd booster dose [33]. After February 1st, 2021, the first and second dose efficacies are reduced to 45% and 75%, respectively, to account for the emergence of the delta variant in India [34]. The above less than 100% efficacy rates mean that vaccinated individuals are not fully protected–they can be reinfected at a rate given by the factor (1 –vaccine efficacy). Note we do not include delays in acquiring of immunity following vaccinations nor do we include differential transmission rates for vaccinated versus unvaccinated individuals. Average vaccination rates estimated from the last 3 weeks of the vaccination data in each country (April 15th -May 6th) were used to simulate into the future. Ramp up of vaccination through the month of simulated lockdowns was carried via a 2-week stepped increase (i.e. By increasing the current rate of approximately 1.1 million daily doses administered in 2-week increments to 1.5x, 2x, 2.5x and 3x this rate) until the vaccination rate that was administered in early April (approximately 3.15 million daily vaccinations) was achieved by the end of the lockdowns. Future scenarios for imposition of additional social restrictions are simply modelled by decreasing the last value estimated for the social mitigation parameter, *d*, by 25%, 50%, or 75%, for durations of time that ranged from 30-, 60- and 90-days.

## Results

### Vaccination and immunity rates

Fig 1 shows the rate of vaccination with at least a single dose carried out in India, model fits to the vaccination data, and predictions of the fractions of the population that had developed immunity or are still susceptible to the virus over time. The results indicate that inclusive of naturally acquired and vaccine-induced immunity, only about 13% of the Indian population was immunized to the virus as of May 5th 2021 before the present wave began to emerge (Fig 2A). While the slow roll out of vaccinations in the country is undoubtedly a key factor underpinning the low level of immunity that had been acquired by the population by that date, it is

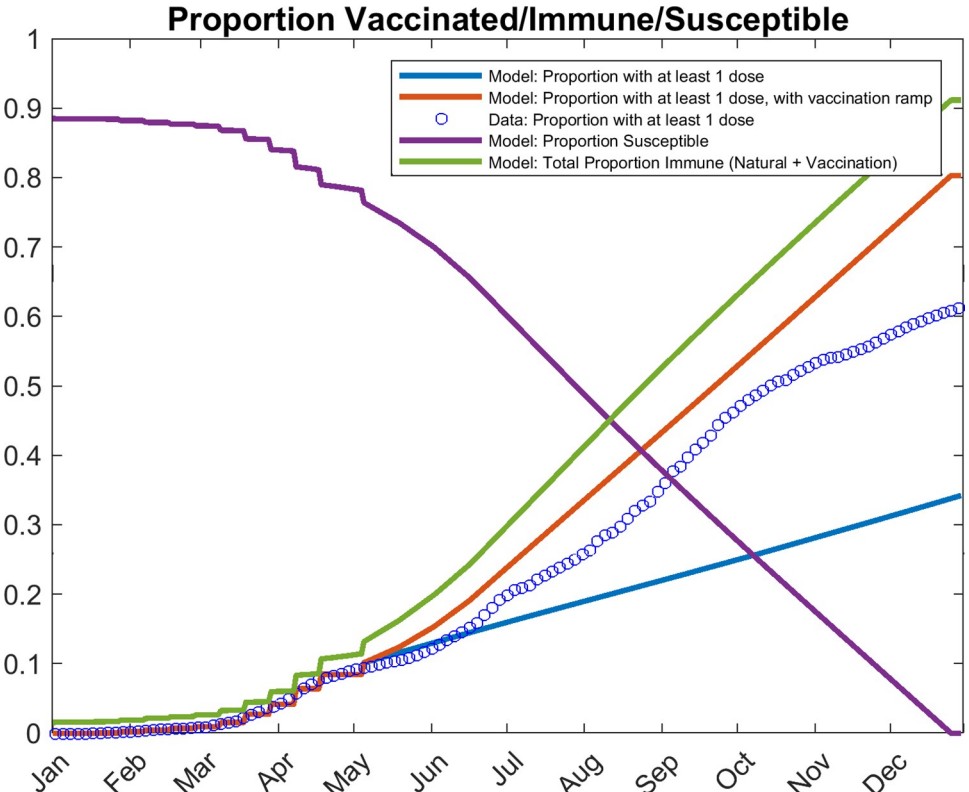

**Fig 1. Median model predictions of proportions vaccinated, susceptible, and immune plotted against reported vaccination data.** Data (open blue circles) are shown every 3 days for clarity. The median model prediction of the proportion of the population with at least one dose of vaccine is given by the blue line. If a biweekly ramp in vaccination ramp is applied after May 5th, the proportion with one dose is given by the red line. The median model predictions of the proportion susceptible and total immune are given by the purple and green curves, respectively. As of May 5th, 2021, approximately 9% of India's population had received at least one dose of vaccine, while only a total of 13% had acquired immunity to the virus.

important to note that this outcome is also the result of the strict lockdown that India imposed successfully to curb the first wave of the pandemic [35]. This means that even though the country had managed to reduce infection and spread drastically during that wave using such a strict lockdown, such suppression of transmission was achieved at the expense of a critically low rate of development of natural immunity. Our results show that this meant that a large fraction of the population (approximately 80%) was left susceptible and vulnerable to infection with the virus once the lockdown was eased from March of 2021 [1,2] (Fig 1).

## Projections of infection cases and hospitalizations

Projections of the resurgence in infections and hospitalizations (total and ICU cases) to the end of 2021 using the models that best-fitted the daily confirmed case data reported for the country to May 5th 2021 are shown in Fig 2. The results indicate that if the level of social mitigation measures to May 5th 2021 and the average vaccinations carried out in the country 21 days prior to this date were to continue without change, median new confirmed and total infectious cases as well as the corresponding total and ICU hospitalizations would increase exponentially resulting in extremely large daily peaks that would occur between mid-July to early August 2021 (Table 1). In terms of daily incidence, the projections show that the pandemic resurgence in the country would have peaked at 840,000 confirmed and 1.73 million

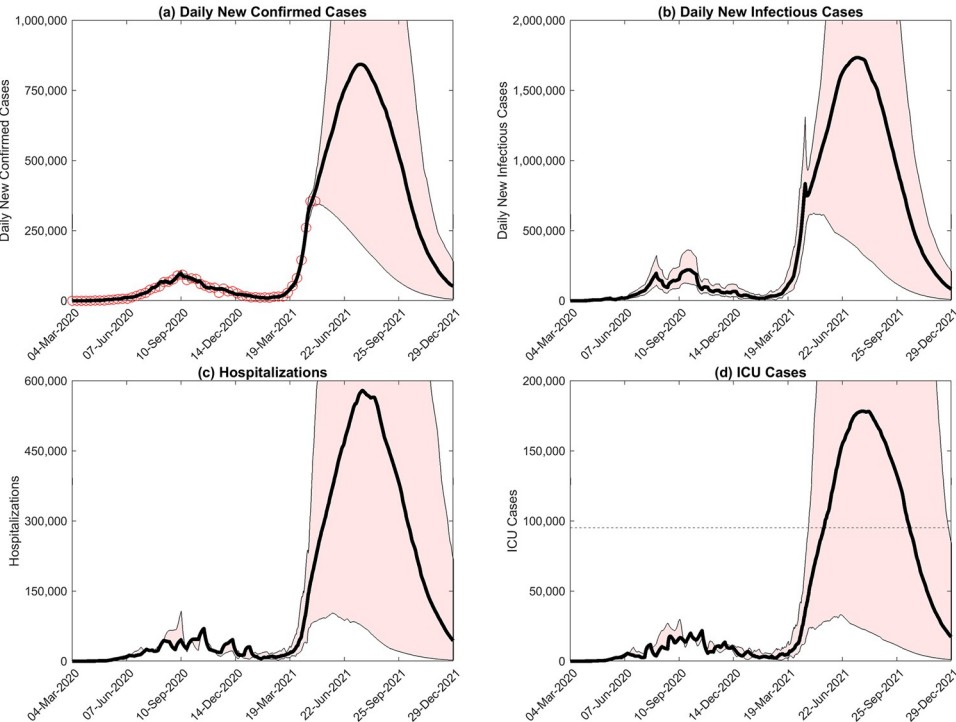

**Fig 2.** Long-term predictions to end of December 2021 of A) daily new confirmed cases, B) daily new total infectious cases, C) hospitalizations, and D) ICU cases, given current estimates of social measures and applying the average vaccination rate of India over 21 days prior to May 5[th] 2021 (approximately 1,470,000 vaccinations per day). Black curves portray the median projections in each case, while 90% confidence intervals are given by the red shaded regions. Confirmed case data is also shown in panel a (red circles, shown every 8 days for clarity).

total infectious new daily median cases during the 3[rd] week of July 2021 if mitigation measures were not imposed immediately (Fig 2A and 2B; Table 1). The corresponding projections for daily total hospitalizations and ICU cases as a result of the second wave are depicted in Fig 2C and 2D, and indicates that without immediate interventions, these cases would also increase exponentially to peaks that will occur towards the middle of July/early August 2021. The predictions show that the resulting peak requirement for ICU beds would vastly surpass the current ICU bed availability (median peak daily need for 180,000 beds versus a capacity of just 95,000 beds country-wide [36]), while the forecasts for total hospitalizations indicate that at peak the expected requirement would consume 31% of all available hospital beds in the country (peak median requirement for 580,000 beds compared to total availability of 1.9 million beds (Table 1)).

The above forecasts for the resurgence of cases and hospitalizations in the country represent, as noted above, the direct outcome of the stringent social restrictions implemented to

**Table 1. Forecasted peaks and dates of peak daily confirmed cases, total infectious cases, hospitalizations, and ICU cases, given current vaccination and social distancing measures.** 90% confidence intervals are given in parentheses.

|  | Peak | Date of Peak |
|---|---|---|
| **Daily Confirmed Cases** | 840,000 (250,000–2,500,000) | July 19[th], 2021 |
| **Daily Total Infectious Cases** | 1,730,000 (370,000–4,900,000) | July 19[th], 2021 |
| **Hospitalizations** | 580,000 (75,000–2,800,000) | July 25[th], 2021 |
| **ICU cases** | 180,000 (22,000–900,000) | August 7[th], 2021 |

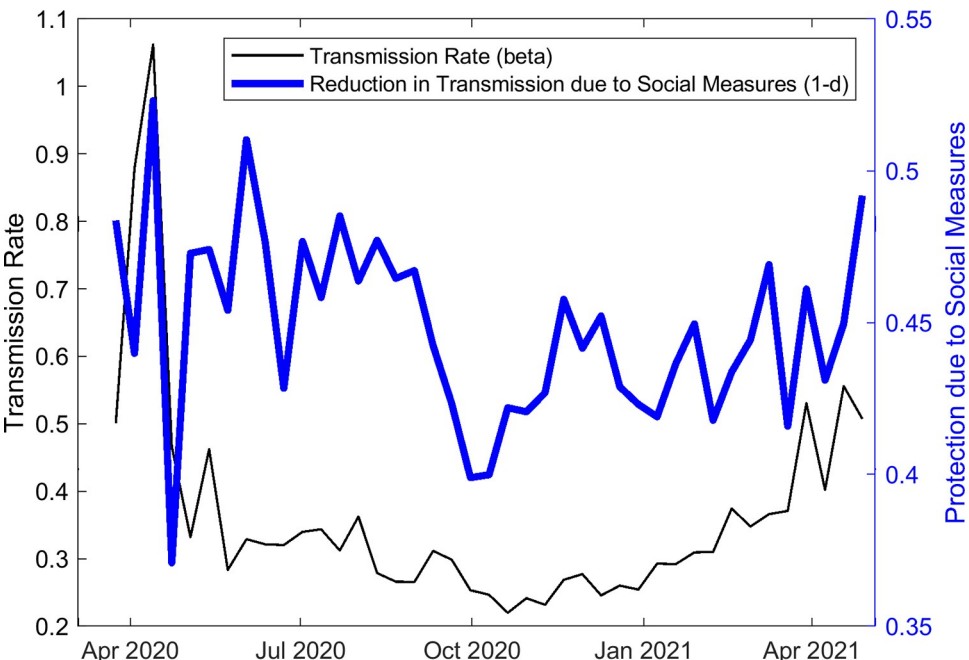

**Fig 3. Model estimations of the median transmission rate (*beta*) and the social distancing parameter (1-*d*) over time.**

curb the first wave coupled with a low vaccination rate [1,2,37] that left approximately 80% of the population still susceptible to infection at the beginning of the resurgence in April/May 2021 (Fig 1). However, the model predictions of the values of social mitigation parameter, *d*, that reduces average population contacts, and *beta*, the transmission rate, shown in Fig 3, indicate that the pandemic resurgence is also related to the easing of social mitigation measures (the term 1-*d* capturing the level of protection against transmission conferred by these measures) and a corresponding but faster increase in the transmission rate across the country, both beginning from October 2020.

## Controlling the resurgent wave

We next evaluated the combination of social mitigation and vaccination measures that may be able to best achieve the control of the unfolding May 2021 wave. We performed this exercise by coupling our best-fit model with scenarios of how combinations of these interventions could be deployed in order to determine their relative impacts on the future course of the intensifying May wave. Model projections of the outcomes of immediate interventions (from May 6th 2021) involving three different levels of social restrictions (a mild 25%, moderate 50%, and strong 75% increase in social protective measures) for durations of between 30 to 90 days either continuing with the current vaccination rate or a 3x ramp up of this rate to the level that was implemented in early April in India, were compared to investigate this topic. The results from these simulations are shown in Fig 4 and Table 2, and indicate clearly that only by immediate imposition of social restrictions could India be able to control the May 2021 wave. However, they also show that important trade-offs between the intensity of social restrictions imposed and its duration with the vaccinate rate can arise from the application of alternative combinations of these interventions. Thus, while the deployment of a shorter duration of social restrictions, eg. the 30-day restriction period, would result in a further resurgence of the

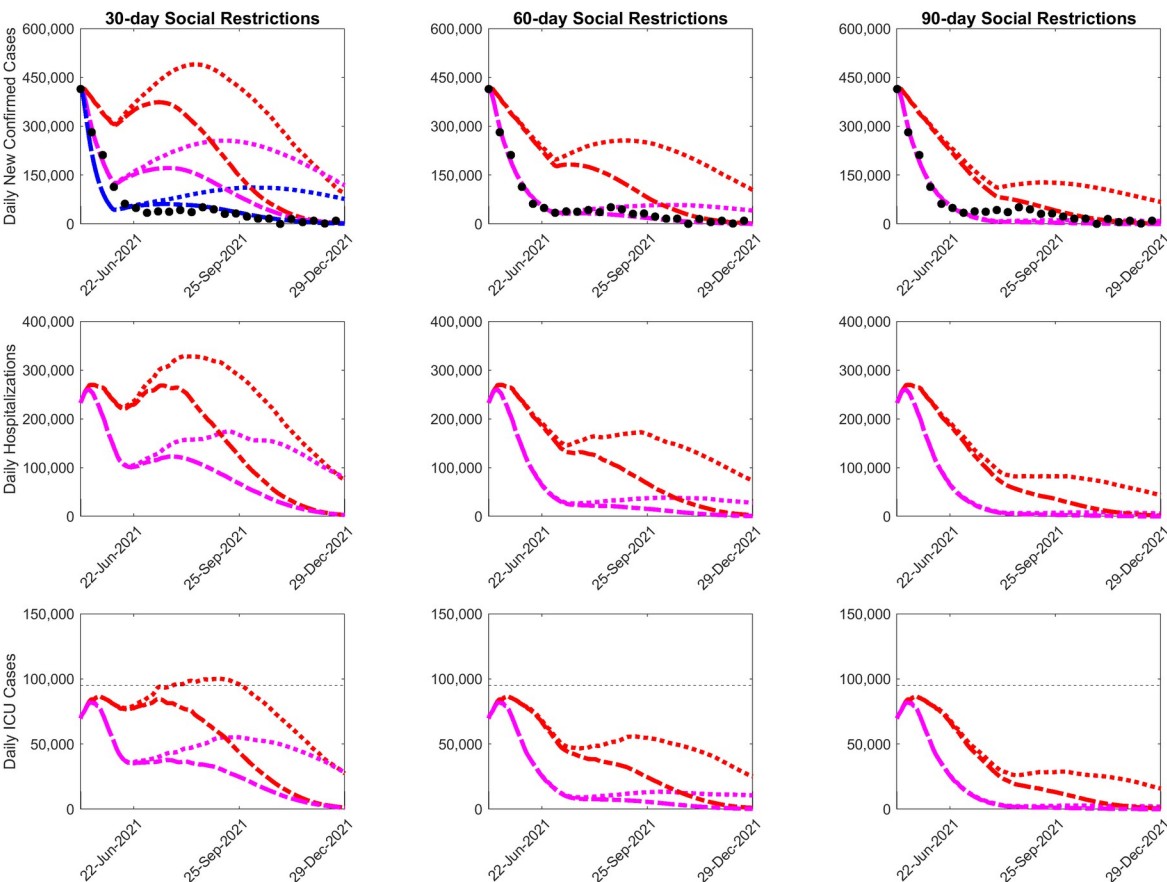

**Fig 4. Long-term median predictions of daily new confirmed cases, hospitalizations, and ICU, given 30-day, 60-day, and 90-day period of social restrictions.** Four scenarios are shown: 1) (Red dotted curve) a 25% increase in social measures along with the vaccination rate prevailing in May 2021, 2) (Red dot-dash curve) a 25% increase in social measures, with a biweekly 50% ramp in first-dose vaccination rate up to 3x the May 2021 rate, 3) (Magenta dotted curve) a 50% increase in social measures with the May 2021 vaccination rate, and 4) (Magenta dot-dash curve) a 50% increase in social measures, with a biweekly 50% ramp in first-dose vaccination rate up to 3x the May rate. On May 5th, 2021, the 21-day average daily first-dose vaccination rate was approximately 1,470,000 vaccinations per day. This rate was increased by 50% every 2 weeks until 3x, which is approximately 4,420,000 vaccinations per day. Confirmed case data are also shown (black circles, shown every 8 days for clarity). A simulation of strong social measures (75% increase in social measures) is shown in blue for the 30-day lockdown scenario.

wave once lifted after the immediate reduction in cases irrespective of the strength of social restrictions imposed and vaccination rates used, such resurgences will be impeded until the wave declines to low levels by end of 2021 if longer periods of social restrictions (either the 60- or 90-day restriction durations) are deployed especially if vaccinations are ramped up (Fig 4). The best intervention for controlling the resurgent wave is that exemplified by the 90-day social restriction scenario, wherein strong control of the May 2021 wave could have been achieved by early August 2021 (ie 3 months after start of the intervention from May 6th 2021) even with a moderately high level of social measures (resulting in 50% reduction in transmission) and continuing with the rate of vaccination observed just prior to May 5ft 2021.

By contrast, the results demonstrate that ramping up the vaccination rate that India managed to administer in April 2021 (approximately 1.1 million doses per day) to at least 3x this rate will be required to curb the late summer resurgences forecasted for the shorter duration social intervention periods investigated here (the 30- and 60-day restriction periods) (Fig 4). This impact of ramped up vaccination particularly when accompanied by a stronger level of

**Table 2. Cumulative deaths due to COVID from May 5<sup>th</sup>, 2021 to December 29, 2021 forecasted for various lockdown and vaccination scenarios.**

| A: Baseline (current social measures and vaccination) | | 1,715,000 | |
|---|---|---|---|
| **B: Intervention Scenarios** | **30-Day social restriction** | **60-Day social restriction** | **90-Day social restriction** |
| **+25% social measures** | 1,260,000 | 844,000 | 615,000 |
| **+25% social measures, biweekly ramp of vaccination to 3x the May 2021 rate** | 789,000 | 539,000 | 445,000 |
| **+50% social measures** | 766,000 | 329,000 | 234,000 |
| **+50% social measures, biweekly ramp of vaccination to 3x the May 2021 rate** | 453,000 | 248,000 | 217,000 |

social restrictions (eg. the 50% increase in social mitigation measures modelled here) will be greater as the duration of social control increases even leading to reduction of the wave to very low levels by the fall of 2021 in the case of both the 60- and 90-day control periods (Fig 4). Ramped up vaccinations with longer durations of social restrictions (60 or 90 days) will further also allow the use of a more moderate (25%) level of social measures for achieving the control of the resurgence. Similar patterns of declines are also predicted for total daily hospitalizations and daily ICU cases, with both these healthcare outcomes declining as lockdown intensity, vaccination rates and social restriction periods increase (Fig 4). Implementing either the 60-day and 90-day lockdown, however, will reduce ICU cases well below India's current ICU bed capacity irrespective of the intensity of social restrictions and vaccination rates modelled (Fig 4).

The cumulative death tolls forecasted for the baseline case (continuation with the social measures and vaccination rates observed just prior to May 5ft 2021) and the different social restriction and vaccination scenarios investigated are shown in Table 2. These results show firstly that by end of December 2021, approximately 1.715 million deaths (and a total of 16 million median new hospitalizations and 6.3 million median ICU cases) may result from the May 2021 resurgence if nothing is done immediately to curb viral transmission across the country. They also show that only by intervening immediately will these staggering outcomes be effectively reduced, with longer and strong levels of social restrictions (i.e. reducing contacts by 50%) coupled with ramped up vaccinations (i.e. to carrying out at least 3.15 million vaccinations per day) able to reduce deaths by up to 77%.

## Assessing scenario forecast credibility

While the results above are based on modelling the impacts of different intervention scenarios on the future paths that may be followed by the May 2021 wave in order to illustrate how alternative interventions may help curb the intensifying outbreak, subsequent case data after the imposition of interventions in India from April/May provide an opportunity to test the ability of our best-fit models to reproduce the trajectory of the wave that was actually realized in the country in the latter half of 2021. S1 Table in Supporting Information summarizes the various social restrictions that were followed by different India states from April/May 2021 in response to the then growing number of cases observed in individual states. While highly variable, in general, across India it would appear that the majority of states went immediately into imposing strong societal lockdowns for at least a month from May onwards (to mid June 2021) before deploying various less stringent social mitigation measures that lasted at least for another month (to end of July and some even into August 2021). India also ramped up its vaccination rate from 1.1 million daily doses carried out in April 2021 to reach 3.134 million doses per day by June 23 2021, a 3x ramp up rate (https://coronavirus.app/tracking/india). Fig 4 compares the predictions of our model for the three social restriction duration scenarios investigated in this paper against case data observed for India from May 2021 to end of December 2021, and show that the 60-day social control scenario employing a moderately high level of social measures (resulting in 50% reduction in transmission on average across the

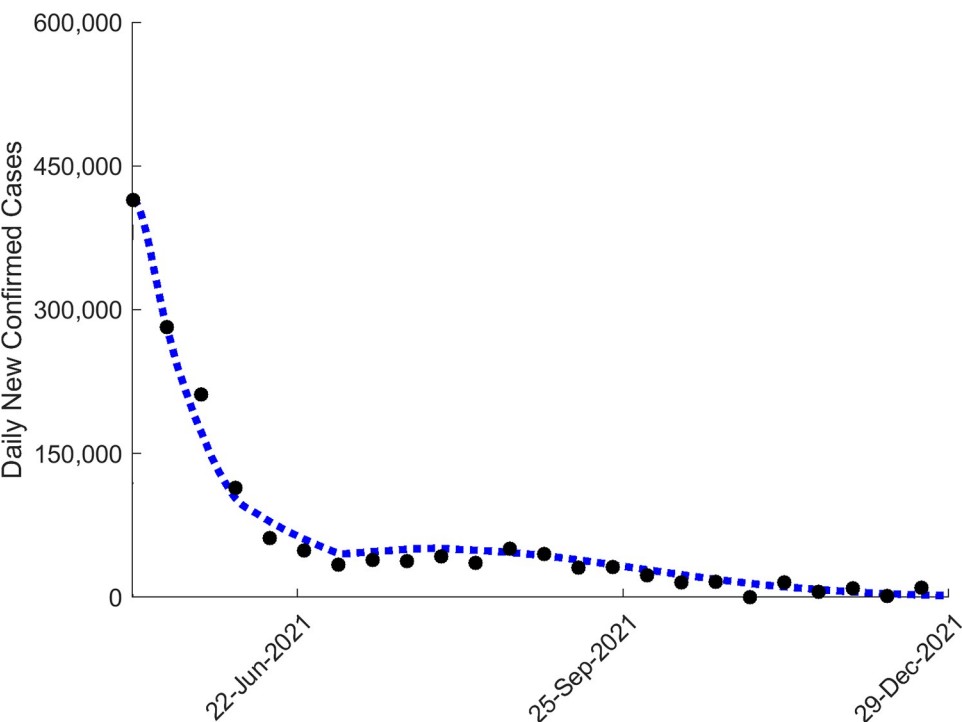

**Fig 5. Long-term median prediction for daily new confirmed cases for the 60-day period of social restrictions with a 3x ramping of vaccinations and a phased imposition of high (60% reduction in transmission) in month one followed by a less stringent moderate (35% reduction in transmission) level of social measures in month two.**

country) and a 6-week ramping up of vaccination from May 5th 2021, which can be taken to approximate the social control applied in India overall, best matched the data.

However, these scenarios applied a constant level of social mitigation levels throughout the durations modelled, and thus may not mimic the phased social interventions that were carried out in India. We modeled the impact of this phased introduction of social measures by considering a scenario in which a high level of social mitigation measures (that reduced transmission by 60%) is imposed during the first month from May 6th 2021 to June 6th 2021 followed by a less stringent phase which reduced average transmission across India by 35% for another month. The median prediction for this scenario in comparison with data is shown in Fig 5; the result indicates that the model describing this scenario is able to faithfully capture the decline in cases observed in India from May 2021 to end of December 2021. These results for the 60-day scenario not only back the decisions made by the most affected Indian states to impose short-term high intensity lockdowns to immediately curb the May 2021 wave followed by phased easing of social restrictions to ensure the gradual decline of new cases in the country as vaccinations ramped up. Our forecasts, however, also show that continuing with social restrictions after a 60-day period given the swift ramping up of vaccinations may not have been strictly necessary to control the May 2021 wave in India, although of course the exact durations and levels of the required social control will likely vary between states depending on their expected wave sizes and vaccination rates.

## Extending the spacing of vaccine doses

We also evaluated the likely impact of India's decision to extend the spacing of the Astra-Zeneca vaccine as a means to get as many people vaccinated (with at least a single dose) in

**Table 3. Forecasted cumulative confirmed cases and hospitalizations from May 5[th] 2021 to December 29, 2021, given 6-week and 12-week dose spacing (after July 1st).** All simulations include a biweekly ramp of vaccination to 3x the rate just prior to May 5[th] 2021.

| | 30-day social restriction | | | |
| --- | --- | --- | --- | --- |
| | 6-Week Dose Spacing | | 12-Week Dose Spacing | |
| | +25% social measures | +50% social measures | +25% social measures | +50% social measures |
| **Daily Confirmed Cases** | 50,139,000 | 27,043,000 | 52,122,000 | 28,267,000 |
| **Hospitalizations** | 6,197,000 | 3,483,000 | 6,410,000 | 3,610,000 |
| | **60-day social restriction** | | | |
| | 6-Week Dose Spacing | | 12-Week Dose Spacing | |
| | +25% social measures | +50% social measures | +25% social measures | +50% social measures |
| **Daily Confirmed Cases** | 33,371,000 | 12,939,000 | 34,559,000 | 13,215,000 |
| **Hospitalizations** | 4,286,000 | 1,814,000 | 4,405,000 | 1,847,000 |
| | **90-day social restriction** | | | |
| | 6-Week Dose Spacing | | 12-Week Dose Spacing | |
| | +25% social measures | +50% social measures | +25% social measures | +50% social measures |
| **Daily Confirmed Cases** | 26,888,000 | 10,820,000 | 27,489,000 | 10,878,000 |
| **Hospitalizations** | 3,569,000 | 1,551,000 | 3,637,000 | 1,558,000 |

order to control the viral resurgence [38]. Table 3 shows the comparative cumulative confirmed cases and hospitalizations predicted for increasing the spacing between vaccine doses from 6 weeks to a 12-week gap for the 3x ramped up vaccination rate (for the period from May 6[th] 2021 to end of the year). These results show that while doubling the gap between doses will increase both the confirmed cases and total hospitalizations, the relative difference will be slight (at most approximately 3% more for the most differential scenario investigated here, viz. the +25% social restriction scenario between the 6-week versus 12-week dosing regimen (Table 3)). The difference in the impact of this increase in the dosing space will decrease as the social control duration is increased and will also be smaller as the intensity of social measures is increased for each control period.

## Discussion

Our results firstly point to the scale to which the resurgent COVID-19 pandemic in India could have grown if measures were not implemented immediately to blunt and curb viral transmission during the exponentially rising phase of the resurgence in May 2021. Of critical concern is the fact that at its peak the projected median daily ICU cases would have completely overrun the available ICU bed capacity in the country, while hospitalization requirements would have consumed approximately 31% of all hospital beds (Fig 2; Table 1). While we project that cumulatively this 2[nd] wave of the pandemic could have resulted in 1.715 million deaths directly, it is clear that overall deaths during the resurgence period could reach even higher levels if hospital capacity to deal with other diseases is diverted to deal with the COVID-19 hospitalizations predicted in this study. Of course, other resource limitations faced by the health system, such as supply of oxygenation units [39], would also further increase the direct death toll from the virus. These conclusions illustrate the potential catastrophic impact that the resurgence might have given rise to if it had gone unchecked, suggesting that an immediate response was required to reduce the epidemic risk was developing in May 2021 in India.

Apart from raising awareness of the potential future growth, size, and duration of the emerging wave and the need for an urgent response, our data-driven modelling also has provided important insights into the merits of different alternative policies for curbing and controlling the viral resurgence as swiftly as possible (Fig 4; Table 2). These overall support the

decisions taken by Indian states to immediately impose a short lockdown while attempting to ramp up vaccinations as the best means for controlling the resurgence of the pandemic. However, the value of using models fitted to real-time data to guide these decisions is highlighted by the results of the intervention simulations presented here (Table 2; Fig 4), which illustrate how forecasts of alternative social restriction and vaccination interventions on the future paths of the May 2021 wave could have been used for identifying the best policy to reduce and suppress the 2nd wave. Thus, while the simulation results show that the longer and more intense the social restrictions implemented after May 2021 are the faster the wave could have been controlled, the best strategy from both the health and economic perspectives would have been to impose a moderately long period (at least for 60 days) of medium level social measures to achieve the effective control of the resurgence. Increasing the lockdown to 90 days could have allowed a more rapid achievement of control even at levels of vaccinations carried out prior to May 2021 (Fig 4), but the strategy will invariably be associated with the high social and economic costs connected with sustaining such a prolonged duration of intervention. Indian states indubitably responded reactively and instinctively to observed trends in cases with some states combining immediate stringent shut-downs that ranged from total societal lockdowns to partial closures of key public spaces to continuing with moderate-level social restrictions (curfews, masking) well into July and August 2021 (ie up to 90 days after May 2021). While these policies were successful in controlling the May 2021 wave across India, our simulations, including the matching of model predictions with observed data (Figs 4 and 5), suggest that India could have achieved control of the wave using a two month period of social measures that could also have been less intense.

A further policy-relevant finding of this study relates to the impact of the government of India's proposal to delay the delivery of the 2nd dose of the Astra Zeneca vaccine, with our simulations showing that extending the gap between the 1st and 2nd doses from 6 weeks to 12 weeks will have only a small deleterious impact in increasing the resulting cases and hospitalizations. This difference will also be reduced with a moderately increased intensity and most effectively by lengthening of the social restriction period (to at least 60 days). These findings thus support the national government's decision to delay the 2nd dosage so that given the constraints of vaccine supply, more people could be vaccinated in the country with at least a first dose to provide at least partial population immunity to the spread of the virus, including to the new delta variant [38].

Overall, these results point firstly to India's overconfidence in March 2021 based on confirmed reported cases during January/February of that year that the country had managed to control the pandemic [1,2,6], and its misreading of these data to mean that a high fraction of the Indian population had acquired immunity to the virus, possibly via cross-immunity gained through exposure to other infections [40]. Our findings show the fallacy of relying on just—indubitably also underreported—confirmed case data to make this assessment given that we estimate the ratio of symptomatic (and presumably confirmed) to total infectious cases, including asymptomatic infections, to be 1- just over 2 (compare Fig 2A and Fig 2B), meaning that a great deal of hidden infection was still ongoing in India when the national government deemed that the country was out of danger and began to open up the economy. Our modelling results also give little credence to the notion that a population can develop high levels of SARS-CoV-2—specific immunity following a stringent social lockdown, particularly like the one practiced in India to curb the first wave, and a slow vaccination rate as that which had been implemented in the country prior to the emergence of the May 2021 wave (Fig 1). Fig 3 also highlights in this regard how estimates of latent variables, such as the temporal changes in the level of social protective measures and in the transmission rate—made possible by sequential model fitting to incoming data—could have warned policy-makers of the imminent risk of

pandemic resurgence across the country. In particular, awareness that decreases in social protection in combination with a corresponding rising transmission rate would likely presage the emergence of a large outbreak, might have significantly tempered the flawed policy decisions to reopen the economy made in March 2021. These considerations imply that relying on and reacting simply to data without consideration of epidemic dynamics could have produced important knowledge-action gaps that hampered appreciation of the developing crisis and the taking of appropriate actions to prevent the April-May 2021 pandemic resurgence by Indian policy makers.

The findings reported here should, as always, be interpreted within study assumptions and limitations. First, it is important to note that our model is data-driven meaning that the results are sensitive to errors in reported data. Although our use of a sequential Monte-Carlo based approach to fit and update parameters of the model with temporal data allows us to reduce prediction errors as best as possible by affording the means to capture rapidly changing, uncertain, transmission conditions, including providing measures of uncertainty in the forecasts (see Table 1), better and timely reporting of case, vaccination and death data at the very least will be needed to improve the reliability of the results reported here. Further, we have used models fitted to only national-level data; this will miss state-level heterogeneities in transmission and intervention effects that did occur in India (S1 Table) although again some measure of this heterogeneity is captured by the uncertainty bounds in our predictions (Table 1). While the impact of the delta variant that first emerged in India in October 2020 [41] is indirectly captured by data-driven model estimates of the transmission rate in this study (Fig 3), data on the proportionate spread of the variant among the reported confirmed cases would have allowed us to simulate its impact more explicitly. Finally, we have assumed that the effectiveness of the Astra-Zeneca vaccine would decrease to 45% following the 1st dose and 75% following the 2nd dose from 67% and 82% respectively after February 1st 2021. This is based on the assumption that the delta variant became significant among the Indian population only after that date. While some studies have reported that efficacy of this vaccine might drop to 30% and 60% after the first and second doses [42], we modelled slightly higher values (halfway between the latter and original efficacy values) to account for the fact that both the original and delta variants of the virus were likely in circulation in the county in May 2021. Again, only genomic surveillance data will allow more reliable modelling of the impact of the delta virus variant. However, dropping the values to the reported delta variant vaccine parameters (30% and 60% after 1st and 2nd doses) did not appreciably affect the results reported here.

We end on a general note arising from our work that pertains to the application of scientific knowledge for the management of large-scale fast-changing societal shocks that are often marked by major uncertainties and incomplete information. While reactive management in combination with past experience can, as was practiced in India, provide a path for controlling the risks arising from these shocks, including those related to fast evolving disease outbreaks or flare-ups, the actions taken using this approach often resemble the application of knowledge in an ad hoc and typically uncoordinated fashion by disparate actors of variable expertise and experience [43,44]. Such a type of emergency management is likely to be less efficient compared to the use of scientific knowledge for assessing both the scale of an intensifying crisis and the best means to achieve its control based on mechanistic links to alternative preventive/mitigative interventions. Our results together with the Indian response to the May 2021 wave show that data-driven epidemic modelling coupled with dynamic forecasts of infection outcomes resulting from carrying out alternative policies can support this shift from following a purely reactive management response based on day-to-day experiential-based policy actions to a more informed proactive plan for bringing about the control of such hazards [43].

## Supporting information

**S1 Table. A summary of the social measures applied by each Indian state from May 2021 – August 2021.**
(DOCX)

## Author Contributions

**Conceptualization:** Edwin Michael.

**Data curation:** Ken Newcomb.

**Formal analysis:** Edwin Michael, Ken Newcomb, Anuj Mubayi.

**Investigation:** Ken Newcomb.

**Methodology:** Ken Newcomb.

**Project administration:** Edwin Michael.

**Supervision:** Edwin Michael.

**Writing – original draft:** Edwin Michael, Ken Newcomb.

**Writing – review & editing:** Edwin Michael, Anuj Mubayi.

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
