## [Decision Letter · Decision Letter 0]

7 Jul 2022

PGPH-D-21-00310

Data-driven model projections and management of the COVID-19 resurgence in India

Dear Dr. Michael,

Thank you for submitting your manuscript to PLOS Global Public Health. After careful consideration, we feel that it has merit but does not fully meet PLOS Global Public Health’s publication criteria as it currently stands. Therefore, we invite you to submit a revised version of the manuscript that addresses the points raised during the review process.

Please respond as much as you can to the comments put forth by both reviewers.

We look forward to receiving your revised manuscript.

Kind regards,

Abram L. Wagner, PhD, MPH

Academic Editor

Journal Requirements:

1. In your ethics statement in the manuscript and in the online submission form, please provide additional information about the patient records used in your retrospective study. Specifically, please ensure that you have discussed whether all data were fully anonymized before you accessed them.

Additional Editor Comments (if provided):

Reviewers' comments:

Reviewer's Responses to Questions

**Comments to the Author**

1. Does this manuscript meet PLOS Global Public Health’s publication criteria? Is the manuscript technically sound, and do the data support the conclusions? The manuscript must describe methodologically and ethically rigorous research with conclusions that are appropriately drawn based on the data presented.

Reviewer #1: No

Reviewer #2: Yes

2. Has the statistical analysis been performed appropriately and rigorously?

Reviewer #1: No

Reviewer #2: Yes

3. Have the authors made all data underlying the findings in their manuscript fully available (please refer to the Data Availability Statement at the start of the manuscript PDF file)?

Reviewer #1: Yes

Reviewer #2: Yes

4. Is the manuscript presented in an intelligible fashion and written in standard English?

Reviewer #1: Yes

Reviewer #2: Yes

5. Review Comments to the Author

Reviewer #1: Line 79 reference missing

Line 90. It would really help the reader understand the importance of this paper if there was a summary, probably here in the introduction of the actual measures implemented in India as a whole, and what happened in individual states. I am having problems matching actual lockdown periods with the modelling. For example, wasn’t a lockdown was imposed in Maharashtra on the 5th April and many states had stated lifting lockdown by mid-June? This does not match the timing in Fig 4.

Line 102. The model makes some critical simplifying assumptions. Whether these makes an overall difference is unclear but should be addressed in the methods or discussion or both.

As far as I can tell from the ODEs at https://github.com/EdwinMichaelLab/COVID-SEIR-India the vaccine is “all or nothing” I.e. there is a certain probability that a vaccinated person is 100% immune from infection or has no immunity at all. This is a critical factor. Data from the UK, and other countries (including India) mainly obtained with the wild type and Beta VOC show that vaccinated people who become infected are significantly less likely to transmit (so the vaccine is leaky) and are much less likely to require hospitalized. This model does not reflect this reality.

Model makes no distinction on the contribution to the community force of infection for people in isolation, in hospital or in an ICU. Again, I can’t tell if this is critical or not, but goes strongly against public health practices. If this was the case, then there would be no point in isolating individuals who become infected.

I don’t see any delays built into the ODE to account for the delays in acquiring immunity following vaccination.

Line 106. Would be clearer if the actual dates for calibrating the model were quoted , Was this 21 Jan 2020 to 5 May 2021 or a shorter period? If it was upto 5th May 2021, then there seems to be a major discrepancy in the fit of the model to the actual data in early May 2021. By 5th May the infection had peaked (i.e. rate of increase was close to zero), whereas the data in Fig. 2,using median values for the model, suggests that the slope on the 5th May was close to maximal. This is critical for Fig. 4 when lockdowns are introduced. Please comment, especially as a model built on sets of differential equations is primarily modelling the slope of the epidemic.

Line 133. Efficacy quoted for wild type is only if the second vaccination at 12 weeks. Second vaccination at 6 weeks gives much less efficacy. See https://doi.org/10.1016/S0140-6736(21)00432-3 for wild type. There is emerging data from the UK that for Delta, at least for non-geriatric populations, AZ efficacy against Delta is similarly reduced if the second vaccination is at 6 weeks. E.g., Fig 4 AZ 50-65 in https://doi.org/10.1101/2021.07.26.21261140

Line 147. Can the authors gives some concrete examples of what would be required to decrease d by 25% or 50%. Without such calibration it is not possible to know whether this level of reduction is realistic.

Line 165. Link to reference 22 is “broken”

Line 169. Fig. 1 Blue line shows data on first vaccines to 5/5/2021 and then the projection to the end of the year. However, this projection is a long way from the actual vaccination coverage that has been achieved to the 19th November. It is currently about 55% single dose – i.e., about twice the projection.

Line 214. It is not clear from the description or the web page listing the ODE how independent estimates are made for d and β. As far as I can tell in the ODE, these parameters only occur as the product i.e., dβ, so I can’t see how these were separately estimated. Is there something I am missing? If so it would be useful to make this clear in the text. Confidence intervals and covariance of d & β?

Line 241. What is shown is a correlation between the model prediction and observed cases. Correlation does not mean causality.

Line 252. 1,470,000 vaccinations per day. Are these first vaccinations, or combined first and second vaccinations? Can these be specified separately.

Line 247. I suggest you do a reality check here on the modelling. During most of April the rate of increase in daily cases was indistinguishable from a true exponential increase with a rate constant of 0.069 per day. Depending on assumptions about the serial interval, that corresponds to an R of about 1.4. Since the % of people vaccinated only had a very minor change over April/May, this suggests that the transmission from person to person (presumably as modelled by the parameter “d”) would need to decrease by about 1 – 1/1.4 = 29% before the epidemic would flatten. So a 25% decreased “d” would be expected to give a slow increase in daily infections shown in Fig. 4 red dotted line for any of the 30, 60 or 90 day lockdown scenarios. On the other hand, a 50% decrease in “d” should lead to decreased infection rates (magenta dotted line) as shown. Is the discrepancy because the epidemic had already peak prior to the lockdowns modelled in Fig 4, in which case the 25% reduction may have been irrelevant?

Line 342. Reference missing

Discussion: This paper and the diagrams are essentially similar to the preprint posted on Research Square on the 3th June. I can’t see any appreciably new modelling since then. I understand PLOS encourages pre-prints but the authors now have the unfortunate situation where their projection can be tested against what controls were put in place, what vaccination levels have been obtained and the number of cases over the period June – November. History has not been kind. As far as I can see, what has actually happened in India does not closely match any of the scenarios/outcome in Fig 4. This largely invalidates at least the first and third goals in the Introduction, lines 85 to 87.

In a revision of the paper, I suggest re-running the model, without further calibration (i.e., use a database locked on the 5th May) against the scenario that actually took place taking into account different situations in different states. Then there can be a real test of the model and a test of the second goal in the Introduction “to facilitate insight into the underlying factors driving the wave of resurgent cases”

Reviewer #2: The manuscript is well written and an interesting piece. Just a few concerns: 1. There is a mix of both past and present tenses, this should be corrected to past since it is a reported event/activity. 2. The abstract section does not clearly summarize the purpose of the study and the results of the study and methodology. More active words would help. 3. In the introduction section of the manuscript, Line 85, it would be more preferable to use "Objectives" i.e we had three objectives rather than "Goal". 4. In the discussion section, Line 404, it would be better to say " Finally we recommend that" rather than saying " we indicate in closing"

6. PLOS authors have the option to publish the peer review history of their article (what does this mean?). If published, this will include your full peer review and any attached files.

**Do you want your identity to be public for this peer review?** For information about this choice, including consent withdrawal, please see our Privacy Policy.

Reviewer #1: No

Reviewer #2: No

---

## [Decision Letter · Decision Letter 1]

7 Sep 2022

PGPH-D-21-00310R1

Data-driven model projections and management of the COVID-19 resurgence in India

Dear Dr. Michael,

Thank you for submitting your manuscript to PLOS Global Public Health. After careful consideration, we feel that it has merit but does not fully meet PLOS Global Public Health’s publication criteria as it currently stands. Therefore, we invite you to submit a revised version of the manuscript that addresses the points raised during the review process.

The reviewer issued a concern that the model doesn't accurately predict the course of the pandemic. I am less concerned about the novelty of the model - another of the reviewer critiques - but I want more background on your projections and how those have actually borne out, and what is different.

We look forward to receiving your revised manuscript.

Kind regards,

Abram L. Wagner, PhD, MPH

Academic Editor

Journal Requirements:

1. In your ethics statement in the manuscript and in the online submission form, please provide additional information about the patient records used in your retrospective study. Specifically, please ensure that you have discussed whether all data were fully anonymized before you accessed them.

Additional Editor Comments (if provided):

Reviewers' comments:

Reviewer's Responses to Questions

**Comments to the Author**

1. If the authors have adequately addressed your comments raised in a previous round of review and you feel that this manuscript is now acceptable for publication, you may indicate that here to bypass the “Comments to the Author” section, enter your conflict of interest statement in the “Confidential to Editor” section, and submit your "Accept" recommendation.

Reviewer #1: (No Response)

2. Does this manuscript meet PLOS Global Public Health’s publication criteria? Is the manuscript technically sound, and do the data support the conclusions? The manuscript must describe methodologically and ethically rigorous research with conclusions that are appropriately drawn based on the data presented.

Reviewer #1: No

3. Has the statistical analysis been performed appropriately and rigorously?

Reviewer #1: No

4. Have the authors made all data underlying the findings in their manuscript fully available (please refer to the Data Availability Statement at the start of the manuscript PDF file)?

Reviewer #1: Yes

5. Is the manuscript presented in an intelligible fashion and written in standard English?

Reviewer #1: Yes

6. Review Comments to the Author

Reviewer #1: 

I’ve been through the revised MS and I note that many of the suggestions have been adopted.

Thank you for that.

However, there are still serious problems with this MS. Not withstanding the comments by the

Authors on p 53 of the PDF, the model does NOT predict the course of the epidemic in India.

As shown in the plots of actual cases in the graphs on pages 49 and 53 of the PDF, (and for a more

extensive view see https://ourworldindata.org/coronavirus ), cases continued to fall after most

states lifted restrictions on the 15th June 2021, reaching very low numbers at the end of December

2021.

The model suggests that the restrictions imposed around the 5th May decreased “d” by about 50%,

but then regardless of the assumptions about the rates of vaccination, there should have been an

immediate and substantial increase in cases following the lifting of restrictions. This did not occur.

Cases continued to fall with NO sign of any inflection in the course of the epidemic when the

controls were lifted.

Clearly the model fails this critical test and until or unless the model is reformulated to get a better

match, then I am afraid I can’t trust any of the conclusions.

On the assumption that the model can be re-worked to give something that matches what actually

happened, there are some other points that authors may like to address. However, I was stopped by

the lack of match between the model and what happened after June 15, and I have not been though

all of the MS in detail again. There may be more points I have missed.

1. To understand this model, it is critical to know what happened with regard to restrictions in

India. There is still no where near enough detail in the Introduction to enable a reader to

match the modelling with what happened “on the ground”. For example, I don’t see

anywhere in this MS that says most states lifted their restrictions around June 15th 2021.

2. The aims need to be revised. It is no longer relevant “to project the size and likely future

paths of the viral resurgence in India” . We know what happened. It may be appropriate to

expand on the second aim, to understand the relative importance of the various factors that

did result in the control of the May 2021 wave of SARS-CoV-2 infection. The discussion

likewise need to consider how well the model predicted outcomes as well as explain the

relative importance of different measures

3. As pointed out by the other reviewer, there are still more places where the tense could be

modified. E.g., in the figure legends.

4. Sorry to be a bit slow, but I’ve read the authors reply but still don’t understand how they

separately estimate β and d. As pointed out before they always occur as the product in

their ODE. Did the authors do this by estimating β separately first? On their github site they

define β = R0 * γ. There are multiple different γ depending on the path though the SEIR

model, but most paths have a γ of ~0.23. With a β of 0.325 that would make R0 about 1.4.

That is impossibly low. (And as far as I can tell, it has to really be R0, as Rt, the ongoing R, is a

function of d and natural and vaccine induced efficacy, so to use Rt to calculate β gets to be

a circular argument again) The original estimate of R0 for the Wuhan variant is about 4 and

for Delta was a bit higher, in the range of 6-8 so there seems to be major problem here with

the parameterization. (Incidentally and presumably not important for this paper, but on the

authors’ github site, the efficacies following single or double vaccinations are apparently

reversed).

7. PLOS authors have the option to publish the peer review history of their article (what does this mean?). If published, this will include your full peer review and any attached files.

**Do you want your identity to be public for this peer review?** For information about this choice, including consent withdrawal, please see our Privacy Policy.

Reviewer #1: No

---

## [Editor Report · Decision Letter 2]

18 Nov 2022

Data-driven scenario-based model projections and management of the May 2021 COVID-19 resurgence in India

PGPH-D-21-00310R2

Dear Prof. Michael,

We are pleased to inform you that your manuscript 'Data-driven scenario-based model projections and management of the May 2021 COVID-19 resurgence in India' has been provisionally accepted for publication in PLOS Global Public Health.

Best regards,

Abram L. Wagner, PhD, MPH

Academic Editor

Minor editorial comment: line 173 states: "Average vaccination rates estimated from the last 3 weeks of the vaccination data in each country (April 15th -May 6th 173 ) were used to simulate into the future. "

I'm not sure what you meant by "in each country" - is that for countries worldwide? Or do you mean each state? Regardless, you can fix if need be during the proofing process.